# Personalized Risk Assessment of Hepatic Fibrosis after Cholecystectomy in Metabolic-Associated Steatotic Liver Disease: A Machine Learning Approach

**DOI:** 10.3390/jcm12206489

**Published:** 2023-10-12

**Authors:** Miguel Suárez, Raquel Martínez, Ana María Torres, Antonio Ramón, Pilar Blasco, Jorge Mateo

**Affiliations:** 1Gastroenterology Department, Virgen de la Luz Hospital, 16002 Cuenca, Spain; msuarezmatias91@gmail.com (M.S.);; 2Medical Analysis Expert Group, Institute of Technology, Universidad de Castilla-La Mancha, 160071 Cuenca, Spain; 3Medical Analysis Expert Group, Instituto de Investigación Sanitaria de Castilla-La Mancha (IDISCAM), 45071 Toledo, Spain; 4Department of Pharmacy, General University Hospital, 46014 Valencia, Spain

**Keywords:** liver fibrosis, cholecystectomy, adaboost, machine learning

## Abstract

Metabolic Associated Fatty Liver Disease (MASLD) is a condition that is often present in patients with a history of cholecystectomy. This is because both situations share interconnected metabolic pathways. This study aimed to establish a predictive model that allows for the identification of patients at risk of developing hepatic fibrosis following this surgery, with potential implications for surgical decision-making. A retrospective cross-sectional analysis was conducted in four hospitals using a database of 211 patients with MASLD who underwent cholecystectomy. MASLD diagnosis was established through liver biopsy or FibroScan, and non-invasive test scores were included for analysis. Various Machine Learning (ML) methods were employed, with the Adaptive Boosting (Adaboost) system selected to build the predictive model. Platelet level emerged as the most crucial variable in the predictive model, followed by dyslipidemia and type-2 diabetes mellitus. FIB-4 score proved to be the most reliable non-invasive test. The Adaboost algorithm improved the results compared to the other methods, excelling in both accuracy and area under the curve (AUC). Moreover, this system holds promise for implementation in hospitals as a valuable diagnostic support tool. In conclusion, platelet level (<150,000/dL), dyslipidemia, and type-2 diabetes mellitus were identified as primary risk factors for liver fibrosis in MASLD patients following cholecystectomy. FIB-4 score is recommended for decision-making, particularly when the indication for surgery is uncertain. This predictive model offers valuable insights into risk stratification and personalized patient management in post-cholecystectomy MASLD cases.

## 1. Introduction

Metabolic-associated steatotic liver disease (MASLD) is a newly coined term to update the definition of non-alcoholic fatty liver disease (NAFLD). The new term aims to encompass the disease’s heterogeneity and the diversity of patients it affects. It also seeks to remove the negative connotation of the word “alcoholic” and the ambiguity of the term “non” from the patient’s perspective. Moreover, it emphasizes the significance of metabolic dysfunction and the various components of metabolic syndrome [1]. MASLD is the most prevalent chronic liver disease, impacting nearly 30% of the global population, although with geographic variations [2,3]. Besides the cardiovascular risk and liver fibrosis progression, the significance of diagnosing MASLD resides in the potential development of hepatocellular carcinoma without cirrhosis, although the underlying mechanisms remain unclear [4,5].

In the context of cholecystectomy, it is noteworthy as one of the most frequently performed surgical procedures worldwide. Between 150 and 200 procedures are performed per 100,000 inhabitants in Europe and the United States annually [6]. The main indication for cholecystectomy is gallstone disease, in any of its various presentations [7]. Additionally, it is associated with metabolic syndrome and serves as the primary biliary manifestation in patients diagnosed with MASLD [8,9]. While most published studies focus on post-surgery complications [10,11], there is evidence suggesting that cholecystectomy may also have metabolic effects, particularly in patients with NAFLD [12,13]. Although the exact pathophysiological mechanisms are not fully understood, it is thought that cholecystectomy can lead to alterations in the regulation of biliary acids within the enterohepatic circulation, mediated by pathways involving the Farnesoid X receptor (FXR) [14,15] and the Fibroblast Growth Factor 19 (FGF19) [16]. Insulin resistance (IR) also has a substantial impact on this process [17,18]. These changes have the potential to trigger the development of MASLD de novo or contribute to the progression of liver fibrosis.

To address this knowledge gap, machine learning (ML) techniques can prove highly valuable. These methods are currently employed in numerous pathologies for the analysis of variables using various mathematical algorithms, aiming to detect patterns and draw conclusions from these data [19]. Some of the methods employed in the field of medicine include K-Nearest Neighbor (KNN) [20], Bayesian Linear Discriminant Analysis (BLDA) [21], Support Vector Machines (SVM) [22], Decision Tree (DT) [23] and Ensamble [24]. By leveraging these ML techniques, we can gain valuable insights into the potential risk of liver fibrosis following cholecystectomy in patients with MASLD, helping improve diagnostic accuracy and personalized patient management.

Given the evidence of the risk of hepatic fibrosis following cholecystectomy in these patients, this study aims to develop a tool to identify patients who can be candidates for surgery without metabolic consequences. When cholecystectomy is unavoidable, the algorithm will assist in identifying which patients will require monitoring to prevent long-term hepatic adverse effects. It will also help prevent unnecessary surgeries when the indication is uncertain. To achieve this, Machine Learning (ML) techniques will be utilized to analyze this database and assess various variables associated with liver fibrosis [25]. In this study, an algorithm based on Adaptive Boosting (Adaboost) has been proposed due to its scalability and parallel computing capabilities. The results demonstrate the superior performance of this method compared to other ML approaches, accurately classifying liver fibrosis detection in previously diagnosed MASLD patients undergoing cholecystectomy. Hence, this method exhibits significant potential to enhance the diagnostic process and the precision of identifying the stated objective in this research.

The document is structured into distinct sections to present the study’s comprehensive findings. Section 2 outlines the materials utilized in the research and provides a detailed description of the proposed method. Section 3 presents the results obtained from the study. In Section 4 and Section 5, a thorough discussion and conclusion are presented, respectively, where the findings are analyzed in-depth, and the overall results of the study are summarized. These sections aim to provide a clear and cohesive presentation of the research outcomes and their implications.

## 2. Materials and Methods

### 2.1. Study Design and Population

This study analyzes a database compiled by four Mexican hospitals from January 2014 to December 2020, which has been deposited in the Harvard Dataverse [26]. The hospitals that participated in the development of this database were: General Hospital of Mexico “Dr Eduardo Liceaga”, Star Médica Hospital, Central Military Hospital, and Medica Sur Clinic and Foundation. The ethical committee of the latter hospital approved this retrospective multicenter cross-sectional study, ensuring compliance with ethical guidelines for human research.

The data retrieval process was carried out in each hospital to identify eligible patients of both genders who were aged 18 years or older and had a diagnosis of MASLD or steatohepatitis with a history of cholecystectomy. The diagnosis was confirmed through either transient elastography (FibroScan) or liver biopsy. The FibroScan technique was performed by experts at each hospital, employing specific cut-offs validated for each equipment: F0–F1 (<6.2 kPa), F2 (6.2–7.8 kPa), F3 (8.2–12.5 kPa), and F4 (9.5–16.1 kPa) [27]. When results were uncertain between the two stages, the hepatologist in charge of the test determined the staging based on the value closest to each fibrosis group. To simplify, patients were divided into two groups: one group composed of those with no or only mild fibrosis (F0-F1) and a second group consisting of those with significant or advanced fibrosis (F2-F4). The histological classification was performed using the validated scoring system by Kleiner et al. [28].

To ensure clarity and prevent any potential overlap between diseases, all patients who met diagnostic criteria for any other chronic liver disease were excluded from the study. This primarily included individuals with viral hepatitis (primarily hepatitis B and C), high alcohol consumption (more than 4 UBE per day for men or more than 2 UBE per day for women), and patients with levels of transferrin saturation > 50%.

For the study’s analysis, the patients were divided into two distinct groups. The first group consisted of patients who received a diagnosis of MASLD at least 6 months after their cholecystectomy being previously free disease. The second group comprised patients who were diagnosed with MASLD at the time of their cholecystectomy. This categorization aimed to differentiate between those with pre-existing MASLD and those who developed the condition following cholecystectomy.

### 2.2. Data Collection

The data collection can be divided into two groups. The first of these is the analytical variables, which are summarized in Table 1. Laboratory data were collected at the time of MASLD diagnosis or within 30 days of the diagnosis. These laboratory data can be divided into variables related to hematology and coagulation (hemoglobin (g/dL), platelet count (10^3^/µL), the international normalized ratio (INR)); general biochemistry (glucose (mg/dL), and lactate dehydrogenase (LDH) (U/L)); lipid profile (total cholesterol (mg/dL), high-density lipoprotein (HDL) (mg/dL), low-density lipoprotein (LDL) (mg/dL), triglycerides (TG) (mg/dL)) and liver function (albumin (g/dL), total bilirubin (mg/dL), alanine aminotransferase (ALT) (U/L), aspartate aminotransferase (AST) (U/L), alkaline phosphatase (ALP) (U/L), gamma-glutamyl transferase (GGT) (U/L)). Other patient data were also collected, including age, gender, Body Mass Index (BMI) (kg/m^2^), which encompasses height (m) and weight (kg), hypertension, type 2 diabetes mellitus (T2DM), and dyslipidemia (DL).

In addition to biopsy and FibroScan, non-invasive tests (NITs) were integrated into the original database to supplement the decision-making process. The study incorporated AST to Platelet Ratio Index (APRI), Fibrosis-4 (FIB-4) and NAFLD Fibrosis Score (NFS) as they are widely used worldwide for assessing liver fibrosis in MASLD patients [29,30].

### 2.3. Model Development

The primary outcome of this study is to develop a predictive tool based on the Adaboost method to identify patients at risk of developing hepatic fibrosis following cholecystectomy in patients with MASLD. This will aid in assessing the indication for surgery in these patients, potentially avoiding unnecessary surgeries or those with a high risk of long-term metabolic complications. In cases where cholecystectomy is deemed essential, it will help identify patients who require long-term monitoring to prevent potential liver-related complications. As secondary objectives, this study aims to assess the risk factors that may contribute to the development and progression of liver fibrosis in these patients. For the analysis, the Adaboost method was proposed as the primary reference system to develop the predictive model. This decision was based on its scalability, fast execution speed, and overall efficiency [31]. Adaboost is known for its parallel tree reinforcement, which enables rapid and accurate solutions to various data science problems. To assess its performance, several other ML algorithms were also tested, including logistic regression (LR) [32], BLDA [33], SVM [34], DT [35] and KNN [36]. The models were designed using Machine Learning Toolbox and MatLab Statistical (The MathWorks, Natick, MA, USA; MatLab R2023a).

In this study, the ML algorithms were fine-tuned by adjusting their respective hyperparameters. For the SVM method, a Gaussian kernel function was selected, with C = 1.2, sigma = 0.5, numerical tolerance = 0.001, and an iteration limit of 100. BLDA utilized the Bayesian kernel for optimization. In the DT method, the parameters were set as follows: maximum number of splits = 20, learning rate = 0.1, and number of learners = 50.

The KNN algorithm employed the Euclidean distance metric and utilized 25 neighbors for each prediction. As for our proposed method, Adaboost, the hyperparameters were optimized with the following values: eta = 0.2, gamma = 0.3, alpha = 0.6, maximum depth = 7, lambda = 0.3, column sample by tree = 0.7 and maximum delta step = 3. These parameter adjustments aimed to enhance the performance and accuracy of each ML algorithm in predicting liver fibrosis in MASLD patients after cholecystectomy.

Figure 1 illustrates the step-by-step ML process followed in this study. To evaluate the algorithm performance, 5-fold cross-validation was employed. This method involved dividing the patient data into five subsets, where 70% of the patients were used for training, and the remaining 30% were used for testing and validation. Each fold represented a different combination of training and testing sets, ensuring that patient data were utilized exclusively in either the training or testing group to avoid any overlap.

Once the database was fully prepared, the ML methods underwent training and validation. The models were trained using the training data, and their performance was assessed and validated using separate testing data. This process allowed for a robust evaluation of each ML algorithm’s predictive capabilities for identifying liver fibrosis in MASLD patients after cholecystectomy. The proposed system met the two objectives set, on the one hand, to obtain a high accuracy in the predictive model and on the other to obtain the predictor variables.

## 3. Results

Results from the analysis of available data for training and validation aimed to identify the most significant variables in predicting liver fibrosis after cholecystectomy in MASLD patients. The performance of the Adaboost system was compared with other ML methods commonly used in the scientific community.

A total of 407 patients diagnosed with MASLD were identified. Among them, 196 patients were excluded due to the absence of a medical history of cholecystectomy, leaving the remaining patients for analysis.

Figure 2 presents a summary of the importance of the primary variables resulting from the developed predictive model. On the y-axis, the absolute importance of each variable within the predictive model is represented with numerical values. Platelet count emerged as the most crucial variable, followed by dyslipidemia (DL) and type 2 diabetes mellitus (T2DM). Body Mass Index (BMI) and hypertension also exhibited some importance, although to a lesser extent compared to the top variables. Notably, low levels of high-density lipoprotein (HDL) and elevated bilirubin levels were found to be relevant factors as well. Among the NITs studied, FIB-4 achieved the highest accuracy in predicting liver fibrosis.

In Table 2, the Adaboost algorithm outperforms all other proposed methods in terms of specificity, F_1_ score, balanced accuracy and Matthews Correlation Coefficient (MCC). Adaboost achieves a balanced accuracy of 93.53%, which is 9.08% higher than the nearest method, KNN, and a specificity of 93.71%, representing a 9.34% difference in favor of Adaboost compared to KNN. Similar trends are observed across all other evaluation metrics.

Notably, the LR method achieves the lowest classification accuracy with a value of 75.64%, representing a significant difference of 17.52% in favor of the Adaboost system.

These results demonstrate that the Adaboost algorithm provides superior performance and accuracy in classifying liver fibrosis in MASLD patients after cholecystectomy compared to the other ML methods considered in this study.

In Table 3, the Adaboost algorithm shows superior performance in all performance metrics when compared to other proposed methods, including SVM, BLDA, DT, KNN, and LR. Adaboost achieves a recall of 93.32% and an AUC of 0.93, representing an improvement of 8.7% and 9.00% over the closest method, KNN, respectively. The Kappa index attains a value of 83,98% for Adaboost, which is 8.93% higher than KNN.

Comparing the results with other methods, Adaboost significantly outperforms DT, BLDA, SVM, and LR in all parameters evaluated. This demonstrates that Adaboost is the most suitable system for implementing the predictive tool to identify liver fibrosis after cholecystectomy in MASLD patients.

Overall, the superior performance of Adaboost in all metrics makes it the preferred choice for accurate and reliable prediction of liver fibrosis in this context.

In Figure 3, the ROC curves of various ML methods are compared to Adaboost, the proposed method. The curves represent the sensitivity and specificity of each method in predicting liver fibrosis after cholecystectomy in MASLD patients.

The Adaboost method exhibits the largest area under the curve (AUC) of 0.93, indicating its superior performance in accurately predicting liver fibrosis. The KNN method comes next with an AUC of 0.84. The larger AUC for Adaboost signifies that it has a more accurate prediction ability, enabling better identification of MASLD patients at risk of hepatic fibrosis following cholecystectomy.

The results from Figure 3 reinforce the conclusion that Adaboost is the most effective ML method for the prediction of liver fibrosis in this study, providing valuable support in clinical decision-making and patient management.

The radar plot in Figure 4 displays the performance metrics of different ML methods in both the training and test phases. The left side of the figure represents the training phase, while the right side represents the test phase. The area of the circle on the plot indicates the performance of each method, with a larger area indicating better predictive capability.

The Adaboost algorithm demonstrated consistent and similar results in both the training and test phases, indicating that it did not overestimate or underestimate. This suggests that the Adaboost algorithm performed well and had good predictive performance, demonstrating its ability to generalize effectively. On the other hand, the rest of the ML methods exhibited smaller areas, indicating lower reliability for the purpose of the study. These methods may not have performed as effectively in predicting liver fibrosis after cholecystectomy in MASLD patients compared to Adaboost.

In summary, the radar plot highlights the superiority of the Adaboost algorithm in both training and test phases, making it the most suitable and reliable method for predicting liver fibrosis in this study.

## 4. Discussion

MASLD and cholecystectomy are both prevalent conditions in the gastrointestinal tract, and though they are often studied separately, there is evidence suggesting that cholecystectomy could potentially worsen the progression of MASLD [12]. In fact, cholecystectomy has been identified as an independent risk factor for MASLD [37].

Indeed, cholelithiasis is another highly prevalent disease worldwide, affecting approximately 10–20% of the adult population [38]. This phenomenon is not only present in the adult population but also in the pediatric population. Two studies conducted in Canadian and Croatian populations demonstrate an increase in the number of cholecystectomies in the pediatric population due to biliary pathology over the last 20 years [39,40]. This increase correlates with the global rise in obesity, which is particularly concerning in this age group. These data highlight the need for pediatricians and pediatric surgeons to become familiar with these situations, as well as the necessity to develop more aggressive health policies to prevent them.

The Hispanic population exhibits one of the highest rates of NAFLD, estimated at 30%. When focusing on the Mexican population, both for MASLD and cholecystectomy, they exhibit significantly elevated data. On one hand, the estimated prevalence of MASLD is over 45%, one of the highest in the world. [41]. This issue is not limited to adults but is also observed in children and adolescents [42]. This is related to a high prevalence of metabolic conditions, primarily including obesity, T2DM, and metabolic syndrome. Furthermore, the higher presence of the Patatin like phospholipase domain-containing protein 3 (PNPLA3) polymorphism in this population contributes to more advanced degrees of hepatic fibrosis [43]. On the other hand, the incidence of gallbladder disease is higher, leading to a significant number of cholecystectomies performed in this population based on available data [44]. Given the association between gallstone disease and MASLD, the potential metabolic consequences of cholecystectomy become all the more relevant in this context. It emphasizes the importance of identifying patients at risk of liver fibrosis after cholecystectomy to better manage their long-term outcomes.

The relationship between cholecystectomy and MASLD involves several interconnected pathways. The issue is that the pathophysiology of this relationship is not fully understood. One of these factors is the IR, which is also interconnected with metabolic syndrome as it is a component of it [45]. Since the majority of gallstones are composed of cholesterol, these are also considered part of the metabolic syndrome in some studies [46]. IR is also one of the primary risk factors for the development and progression of MASLD. It also contributes to the formation of gallstones by creating lithogenic bile through increased cholesterol in the bile and reduced bile acid (BA) synthesis [47]. Furthermore, it is known that the mucosa of the gallbladder plays a role in the secretion and regulation of insulin. When cholecystectomy is performed, it leads to an increase in insulin resistance, thus contributing to MASLD. Patients who have undergone cholecystectomy also exhibit more lipid metabolism abnormalities [48]. This situation worsens insulin resistance and the patient’s metabolic health, which is associated with a potential deterioration of MASLD and an increased risk of liver fibrosis.

Another pathway linking cholecystectomy and MASLD is mediated by various mechanisms that regulate bile composition. Among these, FGF-19 emerges as one of the most relevant. This protein plays a crucial role in lipid metabolism, particularly in relation to BA. One of its functions is to regulate the absorption of these in the terminal ileum [49]. While the gallbladder is present and stores bile inside, it can cause negative feedback when it comes to bile acid reabsorption. After cholecystectomy, this feedback disappears, causing an increase in FGF19 concentrations and alterations in the enterohepatic circulation of bile acids, thereby exposing the liver to higher amounts of various lipids that may lead to the development of MASLD [50]. These findings were confirmed in animal models, although the exact mechanisms remain incompletely understood [51].

In addition to FGF-19, there are other pathways such as pregnane X receptor (PXR) or G-protein-coupled bile acid receptor-1 (TGR5) that play a role in BA metabolism and are associated with the development and progression of MASLD by influencing regulatory mechanisms of IR and intestinal microbiome homeostasis [52,53]. The complex interplay of these mechanisms, where cholecystectomy reduces the elimination of excess cholesterol and exposes the liver to higher concentrations of BAs, combined with other pathways, can contribute to the onset or progression of liver fibrosis in MASLD patients [37].

The study’s conclusion highlights the importance of platelet levels as the most significant risk factor to consider in MASLD patients for predicting liver fibrosis when cholecystectomy is performed. Low platelet levels could serve as an early indication of advanced liver fibrosis or cirrhosis, which warrants further investigation [54]. Furthermore, patients with poorly controlled LDL and T2DM require special attention, particularly when referring to this first variable. These findings can be explained by the excess cholesterol and bile acids to which the liver is exposed after cholecystectomy. Other important variables, although not at the same level, were high BMI, poorly controlled hypertension, and advanced age.

To aid decision-making, the study included two of the most commonly used NITs that could be calculated with the available data. The inclusion of NITs was essential to detect undiagnosed patients with advanced liver fibrosis, so it can be used as a useful screening tool. FIB-4 emerged as the most reliable and valuable in assisting the decision-making process, especially when there are uncertainties regarding the indication for cholecystectomy Since it is used at the initial stage, it can also be valuable in monitoring these patients to assess fibrosis progression and intervene before hepatic decompensation occurs [55].

To assess the originality of the study, a literature review related to the article’s objectives was conducted. No results were found for the application of ML techniques. Nor were any results found for the definition of MASLD in the reviewed literature. When the search was expanded to include the terms Metabolic-associated fatty liver disease (MAFLD) and NAFLD, results were found, but they were not related to ML techniques. Although some results were contradictory, the majority supported the relationship between cholecystectomy and NAFLD.

Yun et al. conducted a study in 82 patients demonstrating the development of significant hepatic steatosis three months after cholecystectomy [56]. Yue et al. conducted a cross-sectional analysis of data involving 14,750 patients, concluding that the risk of developing NAFLD attributable to cholecystectomy was similar to central obesity but not as significant as metabolic syndrome or insulin resistance [48]. A cohort study conducted by Chang et al., which included nearly 300,000 patients, also demonstrated the association between cholecystectomy and the development of incident NAFLD [57]. Finally, two extensive meta-analyses conducted by Jaruvongvanich et al. [58], with 63,000 patients, and Luo et al. [13], with over 27 million patients (specifically 233,537 NAFLD cases), also concluded that cholecystectomy is a risk factor for the development of NAFLD. These are just some examples from the literature found.

As no studies specifically involving ML were found, this research proceeded to employ various ML methods widely used in the scientific and medical communities to determine the most appropriate approach. The Adaboost system exhibited superior results across all evaluated parameters, with almost all of them exceeding 90%. Adaboost demonstrated its reliability in automatically classifying the study’s objective, and its performance was consistent in both the training and test phases, indicating robust generalizability. Moreover, Adaboost exhibited high scalability and execution speed, making it a valuable tool for decision-making in routine clinical practice [59]. However, logistic regression (LR) showed poorer results compared to the different ML methods utilized in this study. This could be attributed to ML techniques’ superior efficiency and accuracy, particularly when dealing with small sample sizes [59].

It is important to acknowledge certain limitations of this research. The sample was drawn from a Mexican population with higher prevalence rates of MASLD and gallstones compared to other populations [60]. The population also exhibits a higher prevalence of obesity and metabolic conditions, such as dyslipidemia and T2DM [61,62]. Additionally, the presence of the PNPLA3 polymorphism, a gene associated with liver fibrosis severity in NAFLD patients, is more frequent in the Hispanic population [63]. The sample size limitation was addressed by employing ML techniques with hyperparameter optimization during the training phase to achieve statistically significant and robust results [64].

## 5. Conclusions

In conclusion, Adaboost has enabled the development of a useful tool for identifying the primary risk factors for liver fibrosis following cholecystectomy in MASLD patients. These variables were low platelet levels, poorly controlled dyslipidemia, and type 2 diabetes mellitus. Special caution is warranted in cases of elevated BMI and uncontrolled hypertension, as these factors further increase the risk of liver fibrosis. Therefore, in patients diagnosed with MASLD, the indication for cholecystectomy should be carefully evaluated in these circumstances. As an additional variable, the presence of an elevated FIB-4 is also useful when assessing surgery as an NIT for the decision-making process.

Among the various machine learning methods explored, the Adaboost algorithm emerged as the most effective in identifying. Its superior performance and scalability make it a promising tool for assisting in clinical decision-making.

Thanks to this predictive model, it is possible to provide personalized management for these patients, potentially avoiding unnecessary surgeries or situations where the long-term risk of complications outweighs the benefits. Furthermore, identifying patients who require long-term monitoring when surgery is necessary can help prevent fibrosis progression and the development of liver-related complications.

## Figures and Tables

**Figure 1 jcm-12-06489-f001:**
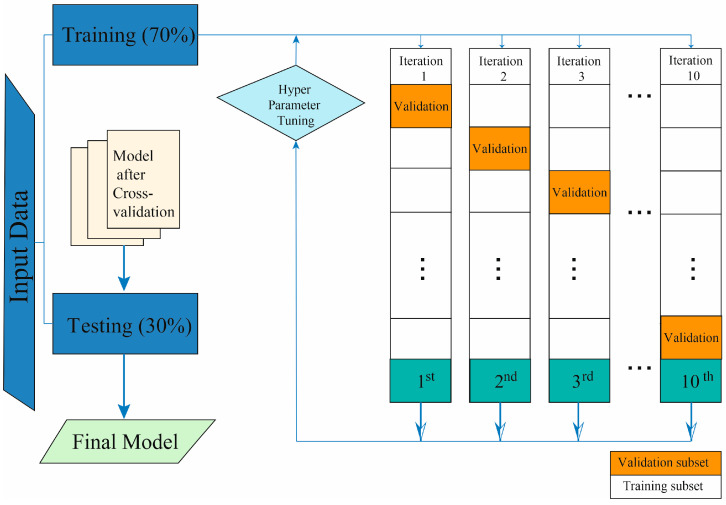
Explanation of the development of the Machine Learning system.

**Figure 2 jcm-12-06489-f002:**
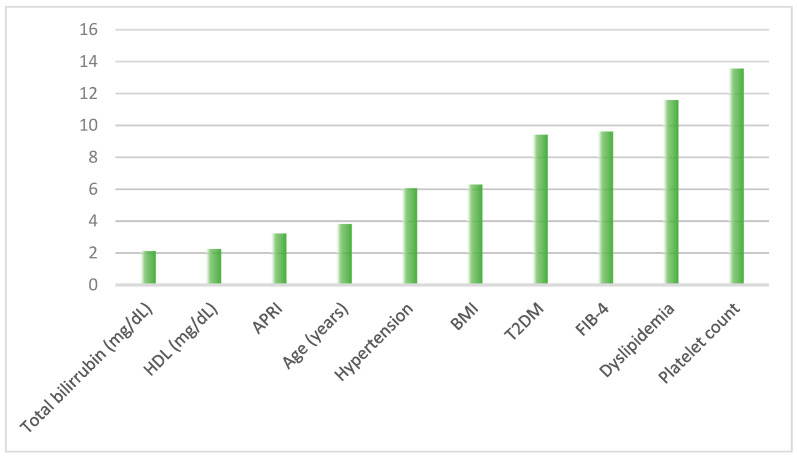
Importance of variables in the predictive model. Abbreviations. HDL: High-Density Lipoprotein cholesterol; APRI: AST to Platelet Ratio Index; BMI: Body Mass Index; T2DM: Type-2 Diabetes Mellitus; FIB-4: Fibrosis-4.

**Figure 3 jcm-12-06489-f003:**
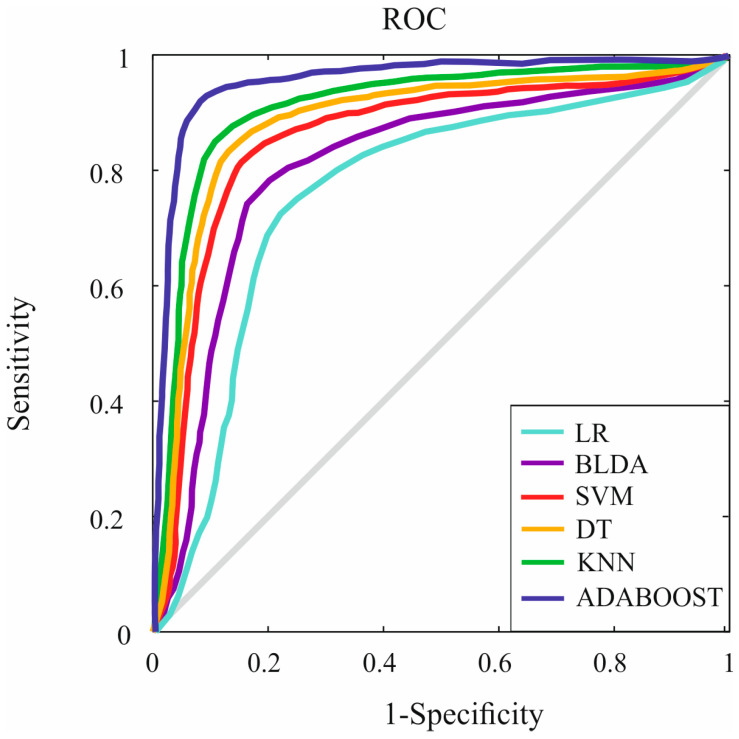
Receiver Operating Characteristic (ROC) Curves for Different ML Methods Compared to Adaboost. Abbreviations. ROC: Receiver Operating Characteristic; BLDA: Bayesian Linear Discriminate Analysis; SVM: Support-Vector Machine; LR: logistic regression; DT: Decision Tree; KNN: K-Nearest Neighbor; Adaboost: Adaptive Boosting.

**Figure 4 jcm-12-06489-f004:**
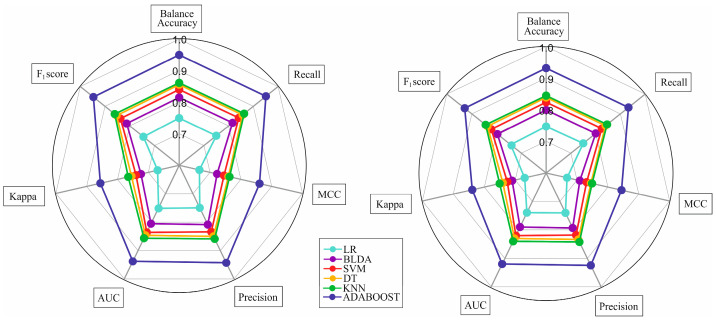
Radar plot comparing different Machine Learning methods in training and test phases. The train phase is represented at the left side of the image, while the test phase is drawn on the right. Abbreviations. BLDA: Bayesian Linear Discriminate Analysis; SVM: Support-Vector Machine; LR: logistic regression; DT: Decision Tree; KNN: K-Nearest Neighbor; Adaboost: Adaptive Boosting; AUC: Area Under the Curve; MCC: Matthews Correlation Coefficient.

**Table 1 jcm-12-06489-t001:** Overview of participant baseline data. Data are also provided by groups. BMI: Body Mass Index; INR: International Normalized Ratio; LDH: lactate dehydrogenase; HDL; High-density lipoprotein; LDL: Low-density lipoprotein; ALT: alanine aminotransferase; AST: aspartate aminotransferase; ALP: alkaline phosphatase; GGT: gamma-glutamyl transferase; APRI: AST to Platelet Ratio Index; FIB-4: Fibrosis-4; NFS: NAFLD Fibrosis Score.

	Global Population (Mean and Standard Deviation)	Patients Diagnosed at least 6 Months after Cholecystectomy. (Mean and Standard Deviation)	Patients Diagnosed at the Moment of Cholecystectomy. (Mean and Standard Deviation)
Sample (*n*)	211	70	141
Age (years)	49.06 ± 15.15	53.15 ± 13.19	47.03 ± 15.69
BMI (Kg/m^2^)	29.19 ± 5.48	30.54 ± 5.37	28.52 ± 5.42
Hemoglobin (g/dL)	13.94 ± 1.88	13.64 ± 1.69	14.09 ± 1.95
Platelet count (10^3^/dL)	256.61 ± 97.24	236.23 ± 110.19	266.72 ± 88.82
INR	1.07 ± 0.17	1.07 ± 0.17	1.08 ± 0.17
Glucose (mg/dL)	116.22 ± 67.76	115.37 ± 54.21	116.64 ± 69.59
LDH (U/L)	244.42 ± 112.03	197.36 ± 101.35	269.76 ± 109.62
Cholesterol (mg/dL)	188.17 ± 52.09	185.89 ± 49.23	189.4 ± 53.71
HDL (mg/dL)	39.6 ± 8.47	43.21 ± 9.49	37.65 ± 7.18
LDL (mg/dL)	102.23 ± 32.54	105.77 ± 32.67	100.33 ± 32.44
Triglycerides (mg/dL)	160.77 ± 77.37	164.56 ± 78.3	158.73 ± 77.09
Albumin (mg/dL)	3.96 ± 0.66	4.08 ± 0.57	3.9 ± 0.69
ALT (U/L)	82.63 ± 143.63	57.51 ± 71.46	95.09 ± 167.20
AST (U/L)	77.81 ± 151.36	77.46 ± 118.58	77.99 ± 165.64
Total bilirubin (mg/dL)	1.4 ± 1.68	1.27 ± 1.65	1.46 ± 1.69
ALP (U/L)	135.97 ± 104.78	121.59 ± 72.23	143.72 ± 117.24
GGT (U/L)	95.31 ± 87.64	107.67 ± 114.14	88.65 ± 68.95
APRI	2.11 ± 1.53	1.08 ± 1.57	0.89 ± 1.52
FIB-4	2.11 ± 3.35	3 ± 4.92	1.66 ± 2.07
NFS	0.42 ± 4.39	0.97 ± 3.73	0.15 ± 4.67

**Table 2 jcm-12-06489-t002:** Comparison of Results—Specificity, F1 Score, Balanced Accuracy, and MCC Kappa. Abbreviations. LR: logistic regression; BLDA: Bayesian Linear Discriminate Analysis; SVM: Support-Vector Machine; DT: Decision Tree; KNN: K-Nearest Neighbor; Adaboost: Adaptive Boosting; MCC: Matthews Correlation Coefficient.

Methods	LR	BLDA	SVM	DT	KNN	Adaboost
Specificity	75.23 ± 0.65	79.82 ± 0.94	82.29 ± 0.77	83.80 ± 0.73	84.37 ± 0.67	93.71 ± 0.48
F1 score	75.58 ± 0.67	79.67 ± 0.92	82.14 ± 0.75	83.68 ± 0.68	84.43 ± 0.64	92.95 ± 0.47
Balanced Accuracy	75.64 ± 0.68	79.92 ± 0.93	82.39 ± 0.78	83.89 ± 0.72	84.45 ± 0.65	93.53 ± 0.51
MCC	66.05 ± 0.67	70.91 ± 0.87	73.10 ± 0.74	74.50 ± 0.68	74.88 ± 0.64	84.69 ± 0.43

**Table 3 jcm-12-06489-t003:** Performance Metrics–Kappa, AUC, DYI, and Recall. Abbreviations. LR: logistic regression; BLDA: Bayesian Linear Discriminate Analysis; SVM: Support-Vector Machine; DT: Decision Tree; KNN: K-Nearest Neighbor; Adaboost: Adaptive Boosting; AUC: Area Under the Curve; DYI: Degenerated Younden’s Index.

Methods	LR	BLDA	SVM	DT	KNN	Adaboost
Kappa	66.54 ± 0.64	71.00 ± 0.93	72.57 ± 0.74	74.02 ± 0.69	75.05 ± 0.63	83.98 ± 0.38
AUC	0.75 ± 0.02	0.79 ± 0.02	0.82 ± 0.02	0.83 ± 0.02	0.84 ± 0.01	0.93 ± 0.01
DYI	75.48 ± 0.69	79.92 ± 0.92	82.39 ± 0.75	83.89 ± 0.71	84.49 ± 0.65	93.45 ± 0.47
Recall	75.86 ± 0.73	80.02 ± 0.91	82.48 ± 0.73	83.99 ± 0.67	84.62 ± 0.62	93.32 ± 0.46

## Data Availability

This database can be found at the Harvard Dataverse.

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
