# Peer review of "Personalized Risk Assessment of Hepatic Fibrosis after Cholecystectomy in Metabolic-Associated Steatotic Liver Disease: A Machine Learning Approach"

_jcm, 2023, doi:10.3390/jcm12206489_

Round 1
Reviewer 1 Report
The authors aimed to develop a predictive tool to identify patients at risk of metabolic-associated steatotic liver disease (MASLD) and liver fibrosis after cholecystectomy, potentially affecting surgical decision-making. They concluded that their predictive model offers valuable insights into risk stratification and personalized patient management in cases of MASLD after cholecystectomy.
I read the study with interest. The study is well written, well designed, and of interest. However, several important questions should be addressed before any positive decisions are made:
1. Abstract – Please include information about the study period and the number of centers included in this study. In addition, among the main results, the authors state that platelet level turned out to be the most important variable in the prediction model, followed by dyslipidemia and type 2 diabetes mellitus. Please insert the calculated values in parentheses next to each variable.
2. Study Period – The authors state that the study was conducted from 2014 to 2020 in four hospitals in Mexico. Please be more specific about the study period and add months.
3. Ethical approval – The authors state that the study was approved by the Ethics and Research Committee for Human Studies of the Medica Sur Clinic and Foundation. Please include a reference number and the date of approval.
4. Outcomes of the study – The primary and secondary outcomes of the study should be clearly stated in the methodology.
5. Table 1 – As I can see from Table 1, all categorical variables were presented as mean and SD. Did the authors test the normality of the data distribution? If not, they should do so and indicate what test they used to do so. Also, data that are not normally distributed should be presented as medians and IQR.
6. Figure 2 – It is unclear what the numbers 0 - 16 on the abscissa represent. Please indicate if it is a percentage, an absolute number, or some other value.
7. Discussion – In their discussion the authors state that cholelithiasis is another highly prevalent disease worldwide, affecting approximately 8-11% of the adult population. I think it would be important to emphasize that gallstone disease has been increasingly diagnosed in the pediatric population as well and the spectrum of pediatric biliary tract disease has been changing. A recent study showed that the number of pediatric cholecystectomies has significantly increased in the last 20 years, as well as the average BMI of the observed population. This probably signifies a correlation between rising obesity rates and an increase in the frequency of symptomatic cholelithiasis in children. Please add a few lines together with the following references. [J Pediatr Surg. 2016; 51(5): 748-52. doi: 10.1016/j.jpedsurg.2016.02.017. and Indian Pediatr. 2019; 56(5): 384-386].
Minor editing of English language required
Author Response
A file has been attached

Reviewer 2 Report
The term Metabolic - Associated Steatotic Liver Disease (MASLD ) includes several factors taking into heterogeneity and diversity of patients suffering from metabolic dysfunction . It is prevalent in patients with Chronic liver diseases . The results in the present manuscript examines and develop a predictive machine learning approach so as to identify patients who are at risk for liver fibrosis after Cholecystectomy . This will help the medical professionals in taking a decision for surgery. Various machine learning methods were tested of which Adaptive Boosting (Adaboost) method gave good base for the predictive model . The Adaboost algorithm gave better outcome in the predictive model . This helps in taking decisions for treating patients when the indications for opting surgery is there when other parameters gave uncertain results. By this predictive model one can provide personalized management of patients with cholecystectomy in MASLD cases.
Clarifications:
Was there any false positive or false negative outcome when assessing the patients by this approach?
Quality of English language used is adequate
Author Response
A file has been attached
